# A Deep Learning Method for Foot Progression Angle Detection in Plantar Pressure Images

**DOI:** 10.3390/s22072786

**Published:** 2022-04-05

**Authors:** Peter Ardhianto, Raden Bagus Reinaldy Subiakto, Chih-Yang Lin, Yih-Kuen Jan, Ben-Yi Liau, Jen-Yung Tsai, Veit Babak Hamun Akbari, Chi-Wen Lung

**Affiliations:** 1Department of Visual Communication Design, Soegijapranata Catholic University, Semarang 50234, Indonesia; peter.ardhianto@unika.ac.id; 2Department of Digital Media Design, Asia University, Taichung 413305, Taiwan; stevet@asia.edu.tw; 3Department of Mathematics, Airlangga University, Surabaya 60115, Indonesia; raden.bagus.reinaldy-2016@fst.unair.ac.id; 4Department of Electrical Engineering, Yuan Ze University, Chung-Li 32003, Taiwan; andrewlin@saturn.yzu.edu.tw; 5Rehabilitation Engineering Lab, University of Illinois at Urbana-Champaign, Champaign, IL 61820, USA; yjan@illinois.edu; 6Kinesiology and Community Health, University of Illinois at Urbana-Champaign, Champaign, IL 61820, USA; 7Computational Science and Engineering, University of Illinois at Urbana-Champaign, Champaign, IL 61820, USA; 8Department of Biomedical Engineering, Hungkuang University, Taichung 433304, Taiwan; byliau@hk.edu.tw; 9Department of Creative Product Design, Asia University, Taichung 413305, Taiwan; 109711569@live.asia.edu.tw

**Keywords:** YOLO, object detection, foot problems, angle parameter, foot clinic

## Abstract

Foot progression angle (FPA) analysis is one of the core methods to detect gait pathologies as basic information to prevent foot injury from excessive in-toeing and out-toeing. Deep learning-based object detection can assist in measuring the FPA through plantar pressure images. This study aims to establish a precision model for determining the FPA. The precision detection of FPA can provide information with in-toeing, out-toeing, and rearfoot kinematics to evaluate the effect of physical therapy programs on knee pain and knee osteoarthritis. We analyzed a total of 1424 plantar images with three different You Only Look Once (YOLO) networks: YOLO v3, v4, and v5x, to obtain a suitable model for FPA detection. YOLOv4 showed higher performance of the profile-box, with average precision in the left foot of 100.00% and the right foot of 99.78%, respectively. Besides, in detecting the foot angle-box, the ground-truth has similar results with YOLOv4 (5.58 ± 0.10° vs. 5.86 ± 0.09°, *p* = 0.013). In contrast, there was a significant difference in FPA between ground-truth vs. YOLOv3 (5.58 ± 0.10° vs. 6.07 ± 0.06°, *p* < 0.001), and ground-truth vs. YOLOv5x (5.58 ± 0.10° vs. 6.75 ± 0.06°, *p* < 0.001). This result implies that deep learning with YOLOv4 can enhance the detection of FPA.

## 1. Introduction

Plantar image analysis is an effective tool for assessing pathological gait and rehabilitation effectiveness widely used in clinical practice [1]. Plantar pressure patterns and distributions, such as foot progression angle (FPA), provide detailed information to evaluate walking abnormalities [2,3,4]. FPA is defined as the angle made between the line of walking progression and the long axis of the foot. FPA represents the foot placement angle of the longitudinal foot axis during gait [5,6,7]. In-toeing and out-toeing, the most common types of FPA deviations, are associated with knee pain and fall risk [8,9]. The average values of in-toeing and out-toeing are established when the FPAs are <0° and >20°, respectively [10,11]. In addition, detecting the FPA can accelerate the rehabilitation process and reduce knee pain [12], such as using ranges of modifications in step width with various amplitudes and gait retraining in everyday walking [13,14] and for in-toeing and out-toeing through proving the effectiveness of medial-wedge insoles and smart shoes [15,16]. However, FPA will determine the gait pathology’s treatment progression, and getting the precise FPA will help the rehabilitation process more efficiently [17,18].

In addition, the impact of FPA may be vital to indicate the plantar pressures changes that can be attributed to chronic disease [19]. The chronic disease is related to knee injury due to excessive toe-in or toe-out [20]. Furthermore, externally rotated FPA and increased medial loading play important roles in flatfoot [21,22]. Moreover, foot placement angle was the best single predictor of total rearfoot motion, and the FPA may be useful to correct atypical rearfoot kinematics [23,24]. Classifying the left foot and the right before measuring the FPA may play an important role in providing information on the postural changes [25]. For example, a decrease in the percentage of body weight on the left heel in asthmatic patients may be related to the postural changes characteristic of asthma [26].

The successful detection of FPA is required to calculate the abnormality angle [27]. The FPA abnormality would not be traced with unexperienced clinical experience, especially in data acquisition and calculation of remote areas [28]. One-dimensional plantar pressure signals [29,30] and two-dimensional plantar pressure images [31] are two methods to capture information on pressure patterns. In addition, the pressure patterns provide detailed information about foot movement [32,33]. Moreover, two-dimensional plantar pressure images can be used to reliably determine the long axis of the foot during walking [34]. Even though plantar pressure software can identify the FPA with a masking algorithm on a foot scan, the masking algorithms present some limitations. The plantar pressure software may not learn the specific classification of foot profiles and detect FPA abnormalities on in-toeing and out-toeing [35,36]. The limitation is an opportunity for deep learning object detection to predict the foot profiles and diagonal FPA in plantar pressure images to get an accurate FPA [37].

This study is intended to examine the effectiveness of deep learning performance on FPA measurements which can be beneficial in excessive in-toeing, or out-toeing foot rotation which alters gait appearance [38]. A more out-toeing gait might reduce pain in patients with knee osteoarthritis. Furthermore, extreme out-toeing reduces patients’ ankle power, potentially mitigating the forces and knee adduction moment, reducing gait speed and efficiency [39]. However, the in-toeing of the FPA induces the reduction of the knee adduction moment. In-toeing is responsible for increasing the knee flexion angle. Therefore, the variation in the in-toeing with the knee flexion angle should be monitored because increasing the knee flexion angle has undesirable effects on knee osteoarthritis progression [36].

Using deep learning for object detection is widely used in biomedical applications [40,41,42]. For example, the deep learning model can identify plantar pressure patterns for early abnormal detection of foot problems [43]. In addition, deep learning-based approaches have presented a state-of-the-art performance in image classification, segmentation, object detection, and tracking tasks [44]. Object detection is suitable to determine where objects are located in a given image and which category each object belongs to [45]. Object detection has become more streamlined, accurate, and faster as the technology has progressed from Region-based Convolutional Neural Network (R-CNN) to Region-based Fully Network (R-FCN). However, these algorithms are region-based [46]. Therefore, image proposals should be created to begin implementing these algorithms. You Only Look Once (YOLO) is not a region-based algorithm and can provide an end-to-end service that makes it more efficient in measuring the FPA. YOLO uses a single neural network design to forecast bounding boxes and class probabilities directly from entire images that may be essential to classifying the left and right foot [47]. Kim et al. found that YOLO outperforms faster than R–CNN, Fast-RCNN, and single-shot detector (SSD) [48]. In addition, YOLO showed good performance in two-dimensional signal detecting medical images [49].

YOLO is a deep learning model commonly used to predict image data such as plantar images [50,51]. YOLO is one of the most powerful and fastest object identification algorithms based on deep learning techniques in providing fast and precise solutions in medical image detection and classification [52,53]. The YOLO networks have several versions that can help accurately detect the FPA. Considering the need for precise results of the FPA, calculations with minimum error values are essential. Therefore, several versions of YOLO networks need to be compared to determine their performance in detecting the FPA in this study. The YOLO network is a one-stage object detection algorithm that can calculate the classification results and position coordinates [54]. Clinical examination of the FPA by the human eye was beneficial to evaluate the in-toeing and out-toeing that related to the basis of postural information [18]. However, evaluating the in-toeing and out-toeing is essential for knee pain information and provides information on the knee pain rehabilitation effect [20]. In addition, changes in FPA affect rearfoot eversion of rearfoot kinematics normalization [55]. This study uses deep learning in object detection for FPA object localization coordinates. Deep learning may improve precision from reported clinical screening results and human accuracies by 10–27% [56]. The precision detection of the FPA can provide information with in-toeing [38], out-toeing [57], and rearfoot kinematics [55] to evaluate the effect of physical therapy programs on knee pain and knee osteoarthritis [5].

## 2. Materials and Methods

Data used to prepare this article were obtained from the AIdea platform provided by Industrial Technology Research Institute (ITRI) of Taiwan (https://aidea-web.tw, accessed on 21 February 2021). This study used 1424 plantar pressure images as datasets, with each image of 120 pixels × 400 pixels. A professional data annotator from the data provider labeled the dataset to classify the foot axis point coordinates in the plantar pressure dataset. The image data were divided into a training set with 900 images, a validation set with 100 images, and a prediction test with 424 images. The labeled prediction test images were used as the ground-truth dataset in this study. However, the ground-truth dataset only provided the front and rear points of the foot axis in pixel coordinates.

Furthermore, the FPA could be calculated using the arctangent formula. All calculations were performed using computer equipment with the following hardware: Core I7-10700 CPU, 32 GB RAM, NVIDIA RTX 3080 10 Gb. This study was reported according to STROBE guideline recommendations [58] for reporting observational studies that were applied during study design, training, validation, and reporting of the prediction model.

YOLO is a state-of-the-art deep learning framework for real-time object recognition. YOLO supports real-time object detection significantly faster than earlier detection networks [50]. This model can run at various resolutions, ensuring both speed and precision, which can be beneficial in measuring the FPA. YOLOv3 became one of the state-of-the-art object detection algorithms [59]. Instead of utilizing mean square error to calculate the classification loss, YOLOv3 uses multi-label classification and binary cross-entropy loss for each label. YOLOv3’s backbone is DarkNet-53, which replaces DarkNet-19 as a new feature extractor. The entire DarkNet-53 network is a chain of many blocks with some strides and 2 convolution layers in between to decrease dimension. Each block has a bottleneck structure of 1 × 1, followed by 3 × 3 filters with skip connections [60]. Alexey has introduced YOLOv4, the next version of YOLOv3, which runs twice as quickly as EfficientDet while providing equivalent performance [61]. Rather than using darknet-53 layers for feature extraction, YOLOv4 uses a modified version of CSPdarknet-53 as a backbone, with cross-stage-partial connections (CSP) employed to split the feature extraction connection into two pieces [62]. Instead of the leaky ReLU function used in YOLOv3 and YOLOv4-tiny, the Mish activation function is utilized in the YOLOv4. YOLOv5 was initially uploaded on GitHub in May 2020, and the maintainer gave the network the name YOLOv5 to avoid confusion with the previous release of YOLOv4 [63]. Implementing the state-of-the-art for deep learning networks, such as activation functions and data augmentation, and the usage of CSPNet as its backbone, are the key new features and enhancements in YOLOv5 [64]. This study used YOLOv3, YOLOv4, and YOLOv5 for measuring the FPA.

The training images were inserted into the YOLO model and processed for training purposes. The information of the predicted bounding boxes could be obtained based on the anchor boxes in the YOLO model. This study compared three different versions, i.e., YOLOv3, YOLOv4, and YOLOv5x, which solved object detection efficiently and straightforwardly [65]. The model’s hyperparameters were as follows: The batch size and mini-batch size were 16 and 4, respectively; the momentum and weight decay were 0.9 and 0.0005, respectively; the initial learning rate was 0.001; the epoch model was 300. The detectors were based on Python 3.7.6, PyTorch 1.7.0 (used in YOLOv5x models), and the Darknet framework (used in YOLOv3 and YOLOv4 models) Windows 10.

### 2.1. Regular FPA Detection Procedure

We conducted five steps to get the FPA (Figure 1) from the data training into calculating the angles. First, we needed to determine the foot profile because the diagonal FPA direction of the left and right foot was different. Second, we trained the diagonal FPA using a bounding box to get the angle-box. The box itself has four corner points in its detection. Third, detecting four angle-box corner points in the diagonal FPA requires acquiring two points (front and rear foot axis points) selected according to the left foot or right foot profiles. Fourth, we used the diagonal FPA to calculate the angle of the FPA using the arctangent formula. Fifth, to confirm our two-foot axis point coordinate predictions, we checked the distance between the predicted and ground-truth points.

#### 2.1.1. Determine the Left and Right Foot Profiles Categories

The first training section labeled the foot profile regarding the left or right position using the bounding box in the dataset (Figure 2A). Furthermore, we input datasets labeled to three different YOLO models, i.e., YOLOv3, v4, and v5x (Figure 2B,C). For the prediction test section, we used 424 images to get the foot profile-box of the left and right feet (Figure 2D). A foot profile-box was used to determine the left foot or right foot position since detecting the foot profiles essential for the differentiation direction of the FPA.

#### 2.1.2. Angle-Box

We used a bounding-box to get the diagonal FPA regarding the angle-box prediction (Figure 3A). In the training section, we input the data labeled by a professional data annotator (Figure 3B) and used the three versions of YOLO models, namely v3, v4, and v5x (Figure 3C). We tested 424 images to get the angle-box prediction and determine the points acquisition based on the foot profile-box (Figure 3D). We used the diagonal FPA on the top left and bottom right for the left foot (Figure 3E), while the right foot diagonal FPA was used on the top right and bottom left.

#### 2.1.3. Point Benchmark Acquisition

After getting the angle-box, the foot axis points were used to get the distance between the ground-truth and three YOLO models (Figure 4). In addition, the YOLO models record four corner coordinates of the angle-box prediction by converting the YOLO coordinates (***x***, ***y***, ***w***, ***h***) into pixel coordinate prediction (***P*_1*x*_***, **P*****_1*y*_***, **P*****_2*x*_***, **P*****_2*y*_**) [66]. In detail, the horizontal value in the front foot axis point was calculated in Equation (1). Next, the horizontal value in the rear foot axis point was calculated in Equation (2). Then, the vertical value in the front axis point was calculated using Equation (3). Finally, the vertical value in the rear axis point was calculated using Equation (4).(1)$$P_{1x}=\frac{x-w}{2}W$$
(2)$$P_{2x}=\frac{x+w}{2}W$$(3)$$P_{1y}=\frac{y-h}{2}H$$(4)$$P_{2y}=\frac{y+h}{2}H$$where ***P*_1*x*_**, ***P*****_1*y*_** represent the front foot axis point coordinates and ***P*_2*x*_**, ***P*_2*y*_** represent the rear of the foot axis point coordinates. The center of the box coordinates is ***x*** and ***y***, the width and height of the bounding box are ***w*** and ***h***, width and height of the images are ***W*** and ***H*** (Figure 4). While ***P*_1*x*_** and ***P*_1*y*_** are the top left corner coordinates for the left foot and the top right corner for the right foot. The lower right corner coordinates for the left foot and the lower-left corner for the right foot are ***P*_2*x*_***, **P_2y_***.

#### 2.1.4. Using Diagonal FPA Detection to Get the FPA

After getting the foot axis points in ***P*_1*x*_**, ***P*_1*y*_**, ***P*_2*x*_**, and ***P*_2*y*_**, we used diagonal FPA from ***P*_1*x*_** and ***P*****_1*y*_** to ***P*****_2*x*_** and ***P*****_2*y*_** to get the FPA results. Then, we calculated the FPA using the arctangent formula [67] to get the angle of the ***A*_1_** and ***A*_2_** (Figure 5). For the calculation, we used Equation (5).(5)$$\theta ={tan}^{-1}\left(\frac{A_{2}}{A_{1}}\right)$$where ***θ*** is the angle in the degree of FPA in each image, the *θ* will be used to differentiate between the ground-truth and the three YOLO prediction results. For example, ***A*_1_** is the height of the angle-box, and ***A_3_*** is the diagonal FPA of the angle-box. The least angles differentiation will conclude the suitable model of YOLO versions in this study.

#### 2.1.5. Measure the Distance

Confirming the foot axis’s front or rear points can affect the diagonal FPA. This study used the two-point distance formula [68] for each image’s ground-truth coordinates and YOLOv3, YOLOv4, and YOLOv5x coordinates values. We calculated the distance of ***G_i_*** (i.e., ***G*_1_** and ***G*_2_**) and ***P_i_*** (i.e., ***P*_1_** and ***P*_2_**) (Figure 4) by Equation (6).(6)$$\vec{G_{l}P_{l}}=\sqrt{{\left(G_{ix},-,P_{ix}\right)}^{2}+{\left(G_{iy},-,P_{iy}\right)}^{2}}$$where $\vec{G_{l}P_{l}}$ is the distance value between the “ground-truth diagonal FPA points coordinates” and “YOLO’s diagonal FPA points coordinates”. We conducted this formula three times to get the distance between the ground-truth and the three YOLO models.

### 2.2. Statistical Analysis

After getting all the values of FPA in the ground-truth and three YOLO models in each image, we compared the front and rear of the foot axis points on three YOLO models using a paired *t*-test. The paired *t*-test was used to describe the differences between points, determine which point affected detecting the FPA, and get the angle differentiation between the ground-truth and YOLO models. Finally, we used one-way ANOVA and LSD post hoc at the significance level of 0.01 to describe the significant difference between YOLO models and the ground-truth. The data were processed using SPSS 26 (IBM, Somers, New York, NY, USA).

## 3. Results

### 3.1. Training Results

Average Precision (AP) and Mean Average Precision (mAP) are the most popular metrics used to evaluate object detection models [69]. A high mAP means that the trained model performs well [60]. Average precision (AP) and loss values of YOLOv3, YOLOv4, and YOLOv5x were calculated, as shown in Table 1. For training results of the profile-box, we used the AP to get detailed results of each class of foot profile-box prediction to determine the left and right foot position [70]. For example, YOLOv4 got the precision of the foot profile-box with an AP of 100.00% for the left foot and was the same high average precision similar for the right foot in 99.78% of AP results. For the foot angle-box, we used mAP. Here, the mAP and AP are the same as the mean because there is only one object. Furthermore, the average precision (mAP) of the training for the foot angle-box for YOLOv4 (97.98%) was 14.38% which was higher than YOLOv5x (96.90%) and 11.88% higher than YOLOv3 (86.32%).

### 3.2. FPA Comparison

The total sample data is 424 images, while the usable sample data is 367 images. This was caused by 57 samples having missing values. Missing values occurred because the deep learning model could not recognize the image; the data were excluded from further analysis [71,72]. Compared with the FPA from the ground-truth, three versions of YOLO models were calculated using one-way ANOVA and LSD post hoc to get the angle differentiation. YOLOv4 FPA (5.86 ± 0.09°) did not show any significant difference compared to ground-truth (5.58 ± 0.10°) (Table 2). However, YOLOv3 and YOLOv5x were different compared to the ground-truth (Figure 6).

### 3.3. Distance between Ground-Truth Point and Prediction Point

To confirm the foot axis point, we used paired *t*-test to get the distance differentiation between $\vec{G_{1}P_{1}}$ and $\vec{G_{2}P_{2}}$ in three YOLO models. Furthermore, we used one-way ANOVA and Fisher’s LSD post hoc to get the distance differentiation of three YOLO models on $\vec{G_{1}P_{1}}$ and $\vec{G_{2}P_{2}}$ (Table 3 and Table 4). The results showed that all comparisons were significantly different (Figure 7 and Figure 8). 

To evaluate the FPA results we found in YOLO models predictions, we used an example plantar image to test our results using angle calculations through digital image software (Photoshop CS.5, Adobe Inc., San Jose, CA, USA) by comparing the ground-truth with YOLO prediction results [73]. First, we measured the ground-truth coordinate and got the FPA. Second, we validated our prediction of the foot axis points coordinates and calculated the FPA. As a result, our forecast approached the ground-truth angle (Figure 9).

## 4. Discussion

This study used the profile-box and the angle-box labeling names to get the FPA. The profile-box uses the whole plantar pressure images to determine left and right foot profiles. The angle-box is inside a plantar pressure image from the heel to the metatarsal head without the toe region and is used to predict the FPA.

This study shows the effectiveness of deep learning with a small-scale data test containing 367 plantar images. In the profile-box, the YOLO training results showed that the YOLOv4 model has the highest mAP with 99.89%, the left foot profile gets the AP with 100.00% accuracy and the right foot profile with 99.78%. Furthermore, the YOLO training showed that the YOLOv4 model gets the highest mAP with 97.98% in the angle-box. However, the results of the FPA between the YOLOv4 prediction and ground-truth angle did not significantly differ, indicating that YOLOv4 and the ground-truth have similar results (Figure 6). Besides, the foot axis’s front point may affect the accuracy of detecting the FPA (Figure 7).

Therefore, the YOLO model is suitable for detecting the FPA from plantar pressure images based on object detection. These results may indicate that YOLO can help predict the FPA. In addition, the precision of YOLO models on the FPA may contribute to clinical practice by providing information on in-toeing, out-toeing, and rearfoot kinematics, in evaluating the effect of physical therapy programs on knee pain and knee osteoarthritis.

### 4.1. YOLO Deep Learning Performance

The normal FPA is an out-toeing angle that ranges from 5° to 13° in children [21]. For the adult population, a normal FPA is defined as between 0° and 20° [10]. Our results indicate that the data used in this study was for a normal FPA (Table 2). Our results showed that the FPA was different from the ground-truth (5.58 ± 0.10°) and three YOLO models (v3: 6.07 ± 0.06°, v4: 5.86 ± 0.09°, and v5x 6.75 ± 0.06°) estimated between 1.3° to 1.9°. The YOLO model can detect and estimate the precise FPA direction of the plantar pressure image. Deep learning can also detect and estimate the spinal curve angle of the trunk kinematics and limb. For spinal disorders and deformities object detection, Galbusera et al. showed that deep learning was trained to predict kyphosis angle, lordosis angle, and Cobb angle. The predicted parameters with an automated method resulted in standard estimate errors between 2.7° and 9.5° [74]. Alharbi et al. showed that deep learning object detection was used to automatically measure the scoliosis angle based on X-rays images and the differentiation from results was estimated at 5°–10° [75].

Furthermore, Hernandez et al. predicted lower limb joint angles from inertial measurement units using deep learning for the lower limb detector and got an estimated average of 2.1° between their ground-truth and predicted joint angles [76]. Pei et al. used deep learning to detect hip–knee–ankle angles in X-rays images, comparing the other deep learning model with a calculated angle ratio that had a deviation from the ground-truth estimate of 1.5° [77]. Our results of different FPAs between YOLO prediction angles and ground-truth angles ranged from 1.3° to 1.9°, similar to the results for lower limb areas in other studies. Therefore, the YOLO models is suitable for detecting the FPA from plantar pressure images based on object detection.

### 4.2. YOLOv4 Showed Superior Results

In our results, YOLOv4 showed excellent performance in detecting the FPA based on plantar pressure images with a single-frame task. The reason would be that YOLOv4 had the backbone network modifications, especially in single-frame tasks, and optimized accuracy for object detection based on images [78]. Whereas YOLOv5 is advantageous in the detection based on video with a multi-frame task [64]. For example, Zheng et al. detected concealed cracks using YOLOv3, v4, and v5x with YOLOv4, proving superior prediction based on single-frame tasks [79]. Furthermore, Andhy et al. applied YOLOv4 to detect waste images based on images and precision results with the actual data [62]. Therefore, YOLOv4′s good performance in the FPA of plantar pressure may be due to the single-frame task.

The results of profile-box training showed that YOLOv4 gets 99.78% (right foot) to 100.00% (left foot) AP due to the characteristic of plantar images with one class and one object in an image. By utilizing boundaries from plantar images, the labeling makes it easier for YOLO to detect foot profiles [80]. Our result was similar to the study by Gao et al. facilitating a robotic arm grasping system in nonlinear and non-Gaussian environment detection using labeling objects on the boundary, with a YOLOv4 range of 96.70% to 99.50% AP. Therefore, YOLOv4 was chosen rather than YOLOv3 and YOLOv5 [81].

In addition, the mAP of angle-box was 97.98% in YOLOv4 was lower than the profile-box mAP of 99.89% (left foot 100.00% and right 99.78% AP). The detection of the angle-box may have limitations on prediction due to the position of the angle-box inside the pressure images with similar background color and density from the pressure. Similar background color and density were the problems of detecting a cluster of flowers and detecting eyes, nose, and mouth in the face. Wu et al. detected apple flowers in natural environments. They got the result of 97.31% mAP on YOLOv4, which had a bounding-box in the flowers with a similar background color and density of flower clusters [11]. Dagher et al. predicted that face recognition to detect the eyes, nose, and mouth was more complex than predicting the whole face [82]. It is concluded that YOLO might be good at profile detection.

### 4.3. Foot Profiles Prediction and Foot Axis Points Distance

Specific markers could predict the FPA front and rear point in two small bounding-boxes. However, the two small bounding-boxes in the front and rear were very similar. Therefore, YOLO was not the best performance for similar objects in one image [83]. The low performance is caused by the fact that just two small boxes in the grid are anticipated and only belong to a new class of objects within the same category, resulting in an abnormal aspect ratio and other factors such as low generalization capacity [84]. Due to these reasons, we used one bounding-box, including front and rear points, to get the FPA.

Similar background color and density were problems in the angle-box and may have affected FPA accuracy in object detection. FPA accuracy is based on the two points of the diagonal FPA (front and rear foot axis point). Therefore, the distance between the predicted and ground-truth points is necessary to investigate. The FPA, especially in the front foot axis point between the three YOLO models prediction and the ground-truth (9.23 to 18.25 pixel), was longer than the rear foot axis point (7.34 to 12.80 pixel). Furthermore, the front foot axis point as a density area also has a similar background of pressure to the metatarsal-phalangeal joints bone and near the other bone, affecting the detection of the used plantar pressure images [85,86]. The density and similar background can lead to low performance in predicting the bounding boxes [87]. However, the rear foot axis point is clearer than the front foot axis point. The rear foot axis point has pressure from calcaneus bone, allowing the YOLO model optimum detection with a non-maximum suppression feature [88,89]. In addition, the rear of the foot axis point is around the boundary of the plantar pressure distribution area with minimum density [90]. The results represent that the front foot axis point due to increasing density from metatarsal-phalangeal joints bone and near the other bone may affect detecting the FPA.

### 4.4. Limitation in Diagonal FPA Acquisitions

The main limitation of our study was the analysis dataset without in-toeing data. As we know, in-toeing is a symptom of illness in the FPA and needs further intervention. Even though we did not have the plantar image with the FPA of in-toeing in this study, our standard methods can be used to measure out-toeing. However, in-toeing measurement has required the addition of the “regular-FPA-procedure” in “labeling the foot profile in left and right categories” and “point benchmark acquisition.”

“Labeling the foot profile in left and right categories” needs to be modified into four classifications: “labeling the foot profile in left-in-toeing, left-out-toeing, right-in-toeing, and right-out-toeing categories.” To determine foot profiles associated with in-toeing conditions by labeling plantar pictures, we used YOLO to do the first classification to get the left and right foot profiles of in-toeing conditions such as left-in-toeing and right-in-toeing. The in-toeing foot profiles position may have the other condition to measure the FPA than the out-toeing condition. In out-toeing, the diagonal FPA acquisition is the same as the “regular-FPA-detection-procedure.” In contrast, in-toeing diagonal FPA acquisition is the patient’s normal foot profile (Figure 10). Therefore, it is necessary to classify the foot position before detecting the foot axis points.

Furthermore, “point benchmark acquisition” was based on the angle-box. YOLO can detect the 4-corner coordinates of the angle-box prediction through the converting stage and then acquire the 2-point benchmark referred to as the foot position of the in-toeing foot direction (Figure 10) [91]. The left and right foot profiles of in-toeing determine the front and rear axis points used to get the diagonal FPA used to measure the angle of the FPA [92].

In addition, using more data validation sets over 3500 images may increase YOLO performance [93]. However, the current study using a small-scale validation set under 350 images showed good performance [42]. Therefore, this study used a small-scale validation set using plantar pressure images and achieved a suitable YOLO performance.

## 5. Conclusions

This study proposed three YOLO models for a suitable model for detecting the FPA. YOLOv4 showed superior results in detecting the left and right foot profiles. Deep learning with YOLOv4 has the advantage of improving predictions of the FPA without significant differences from the ground truth. Besides, YOLOv4 has a reliable detection accuracy of FPA from plantar pressure images. The effects of the accuracy of the FPA may be from the front of the FPA point. The precision detection of the FPA can provide information with in-toeing, out-toeing, and rearfoot kinematics, to evaluate the effect of physical therapy programs on knee pain and knee osteoarthritis.

## Figures and Tables

**Figure 1 sensors-22-02786-f001:**
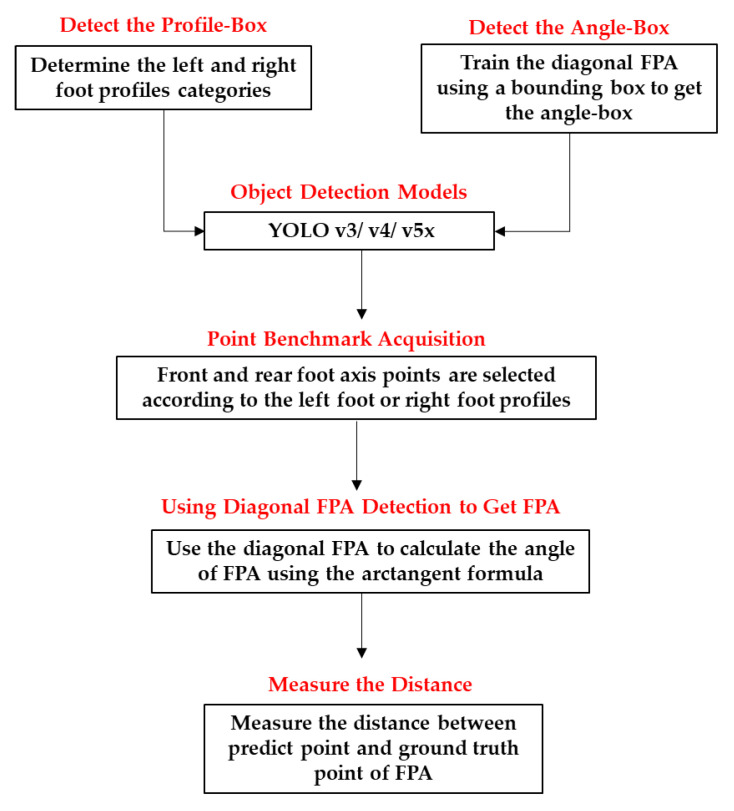
Flowchart illustrating the proposed method of foot progression angle (FPA) detection using YOLOv3, YOLOv4, and YOLOv5x.

**Figure 2 sensors-22-02786-f002:**
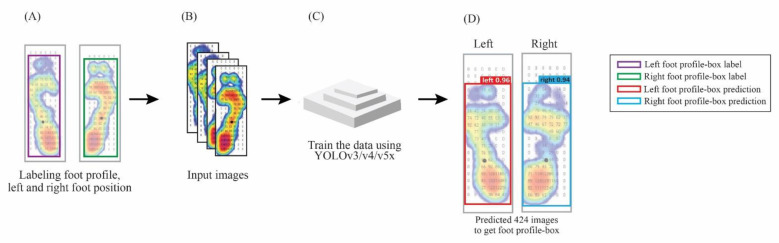
The illustration of profile-box for left and right foot in plantar pressure image detection with the YOLO model. (**A**) Labeling the foot profile in left and right categories using the bounding-box. (**B**) Input the data labeled for training. (**C**) Training the dataset using YOLOv3, v4, and v5x. (**D**) The prediction test used 424 images to get the foot profile-box.

**Figure 3 sensors-22-02786-f003:**
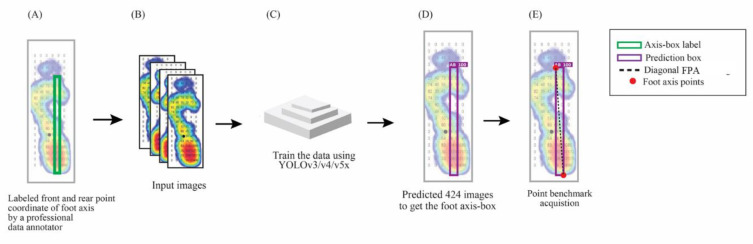
The illustration of angle-box for foot progression angle (FPA) in plantar pressure image with the YOLO model. (**A**) The dataset is labeled by the professional annotator (**B**). Input the data of labeled images. (**C**) Training using YOLOv3, v4, and v5x. (**D**) testing the 424 images to get the foot angle-box. (**E**) Determine the foot axis point acquisition based on the foot profile-box.

**Figure 4 sensors-22-02786-f004:**
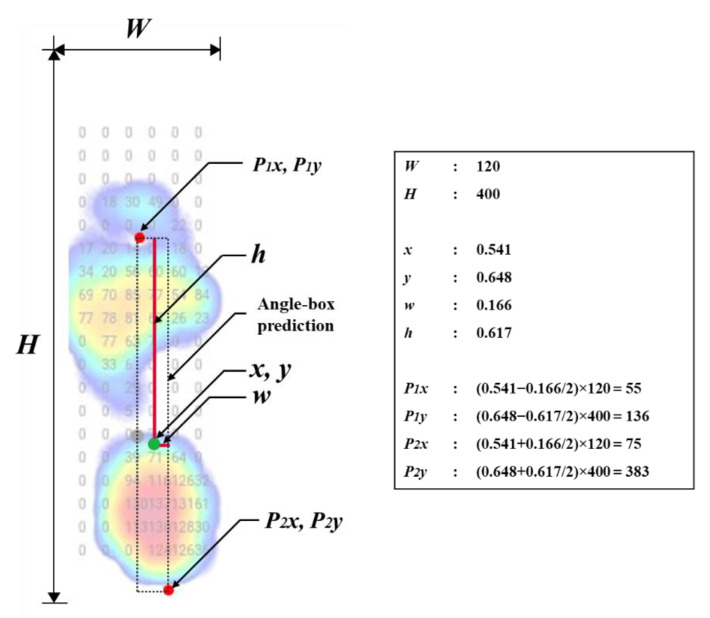
The example of converting YOLO coordinates (i.e., angle-box) into pixel coordinates. YOLO coordinates are ***x***, ***y***, ***w***, ***h***, ***W***, and ***H***. Pixel coordinates ***P*_1*x*_**, ***P*_1*y*_**, ***P*_2*x*_**, ***P*_2*y*_**; ***X*** and ***Y***, coordinates represent the center of the box; ***w***, the width of the bounding-box; ***h***, the height of the bounding-box; ***W***, the width of image; ***H***, the height of images.

**Figure 5 sensors-22-02786-f005:**
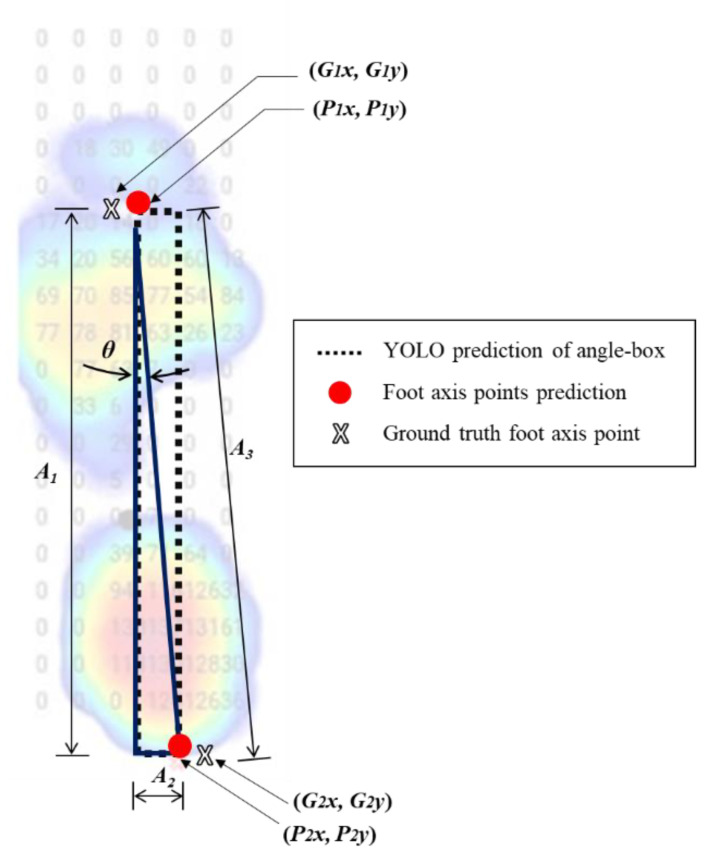
The example of diagonal foot progression angle (FPA) of the angle-box in the YOLO model; ***G*_1_** (***G*_1*x*_** and ***G*****_1*y*_**), Ground-truth for the front foot axis point; ***G*_2_**, ground-truth for the rear foot axis point in (***G*_2*x*_** and ***G*****_2*y*_**); ***P*_1_** (***P*_1*x*_** and ***P*****_1*y*_**), YOLO models prediction for front foot axis point; ***P*_2_** (***P*_2*x*_** and ***P*****_2*y*_**), YOLO models prediction for the rear foot axis point; ***A*_1_**, the height of angle-box; ***A*_2_**, the width of the angle-box; ***A*_3_**, the diagonal FPA of the angle-box; ***θ***, in degrees for FPA.

**Figure 6 sensors-22-02786-f006:**
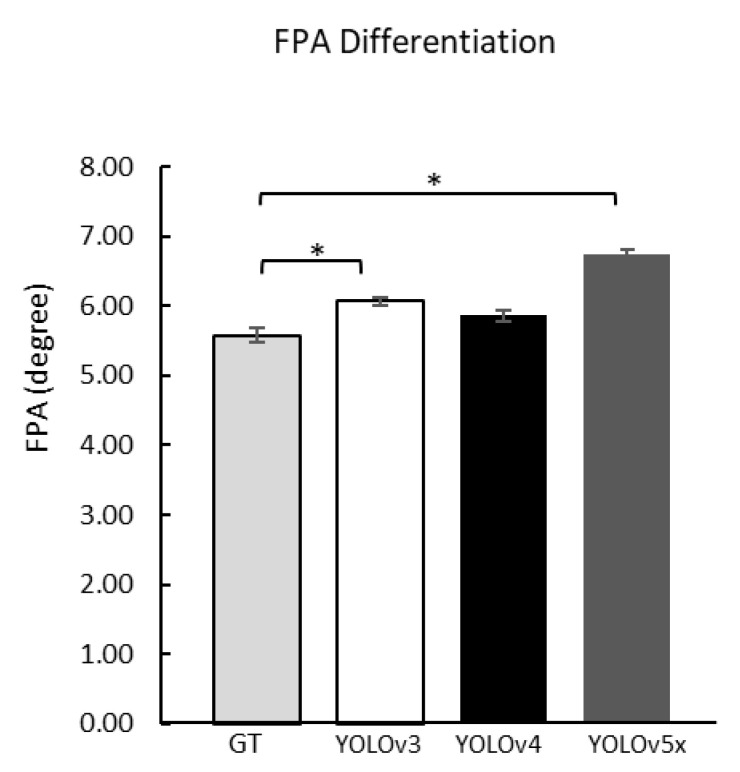
FPA comparison between GT with YOLOv3, v4, and v5x. GT, Ground-truth; *, a significant difference (*p* < 0.01).

**Figure 7 sensors-22-02786-f007:**
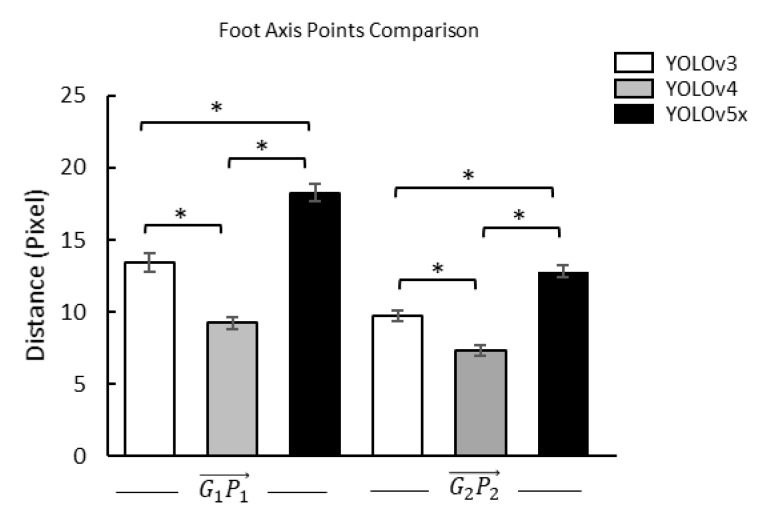
Comparisons of the front ($\vec{G_{1}P_{1}}$) and rear of the foot axis point ($\vec{G_{2}P_{2}}$) of the angle-box on the distances between ground-truth and prediction points in different YOLO models. $\vec{G_{1}P_{1}}$, the distance between the front points of the ground-truth (***G*_1_**) and YOLO model prediction (***P_1_***); $\vec{G_{2}P_{2}}$, the distance between the rear points of the ground-truth (***G_2_***) and YOLO model prediction (***P*_2_**); *, a significant difference (*p* < 0.01).

**Figure 8 sensors-22-02786-f008:**
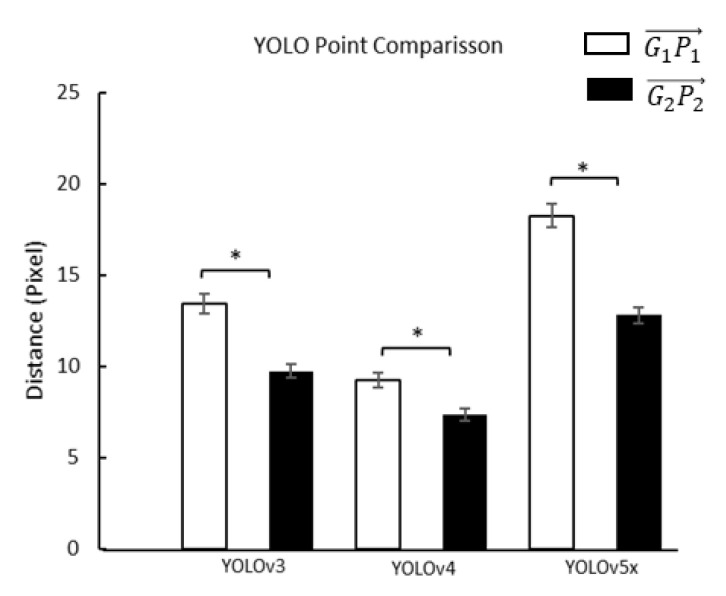
Comparisons of the effect of different YOLO models on the distances between the ground-truth point and prediction point at the front point of the angle-box ($\vec{G_{1}P_{1}}$), and rear points of the angle-box ($\vec{G_{2}P_{2}}$). *, a significant difference (*p* < 0.01).

**Figure 9 sensors-22-02786-f009:**
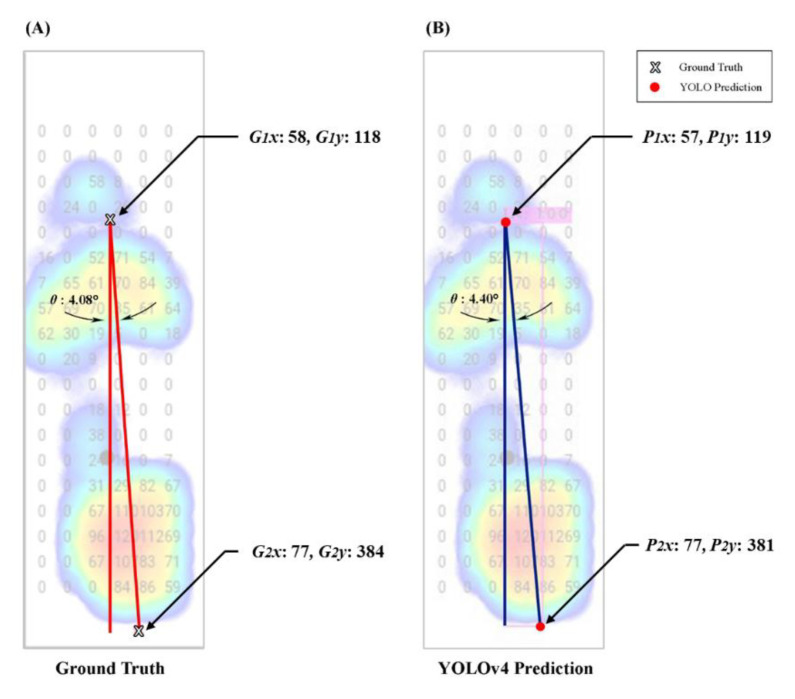
Examples of validation using photoshop software for the left foot profile. (**A**) the ground-truth angle of 4.08° was compared with (**B**) the same images from the YOLOv4 prediction with an angle of 4.40°. Note: ***G*_1*x*_** and ***G*****_1*y*_** are front foot axis points of the ground-truth. ***G*_2*x*_** and ***G*****_2*y*_** are rear foot axis points of the ground-truth. ***P*_1*x*_** and ***P*****_1*y*_** are the front foot axis point of the YOLO model. ***P*_2*x*_** and ***P*****_2*y*_** are the rear foot axis point of the YOLO model.

**Figure 10 sensors-22-02786-f010:**
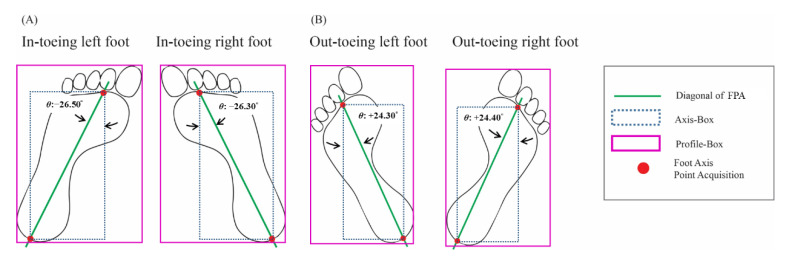
Different foot positions of in-toeing and out-toeing will acquire other foot axis points. (**A**) the method for diagonal FPA acquisition of in-toeing. (**B**) the method for diagonal FPA acquisition of out-toeing.

**Table 1 sensors-22-02786-t001:** YOLOv3, v4, and v5x performance in bounding box training on the foot profile (profile-box) and FPA (angle-box).

Bounding	Parameter	YOLO Version		
Box Type		v3	v4	v5x
Profile-box				
	mAP	86.32%	99.89%	96.90%
	Loss	0.55	0.12	0.00
	Left foot (AP)	92.93%	100.00%	95.80%
	Right foot (AP)	79.70%	99.78%	98.00%
Angle-box				
	mAP	86.01%	97.98%	83.60%
	Loss	1.47	0.53	0.02

Note: mAP, mean average precision; AP, average precision; FPA, foot progression angle.

**Table 2 sensors-22-02786-t002:** One-way ANOVA of FPA comparison between the ground-truth angle and different YOLO versions.

Parameter	Ground-Truth	Model			One-Way	Fisher LSD		
					ANOVA			
	(Mean ± SE)	YOLOv3 (Mean ± SE)	YOLOv4 (Mean ± SE)	YOLOv5x (Mean ± SE)	*p*-Value	GT vs. YOLOv3	GT vs. YOLOv4	GT vs. YOLOv5x.
*θ* (degree)	5.58 ± 0.10	6.07 ± 0.06	5.86 ± 0.09	6.75 ± 0.06	<0.01 *	<0.01 *	0.013	<0.01 *

Note: *θ*, angle in degree; GT, Ground-truth; FPA, foot progression angle; *, a significant difference (*p* < 0.01)

**Table 3 sensors-22-02786-t003:** Effect of different YOLO models on the distance between the ground-truth point and prediction point.

Parameter	YOLO			One-Way	Fisher’s LSD		
				ANOVA	Post Hoc		
	v3 (Mean ± SE)	v4 (Mean ± SE)	v5x (Mean ± SE)	*p*-Value	YOLOv3 vs. YOLOv4	YOLOv3 vs. YOLOv5x	YOLOv4 vs. YOLOv5x
$\vec{G_{1}P_{1}}$ (pixel)	13.41 ± 0.52	9.23 ± 0.39	18.25 ± 0.62	<0.01 *	<0.01 *	<0.01 *	<0.01 *
$\vec{G_{2}P_{2}}$ (pixel)	9.74 ± 0.38	7.34 ± 0.36	12.80 ± 0.43	<0.01 *	<0.01 *	<0.01 *	<0.01 *

Note: $\vec{G_{1}P_{1}}$, the distance between the front points of the ground-truth (***G*_1_**) and YOLO model prediction (***P*_1_**); $\vec{G_{2}P_{2}}$, the distance between the rear points of the ground-truth (***G*_2_**) and YOLO model prediction (***P*_2_**). The value coordinates in this calculation were in the pixel; *, a significant difference (*p* < 0.01).

**Table 4 sensors-22-02786-t004:** Effect of different points of FPA on the distance between the ground-truth point and prediction point.

	Distance		Paired *t*-Test
	$\vec{G_{1}P_{1}}$	$\vec{G_{2}P_{2}}$	
Model	(Mean ± SE)	(Mean ± SE)	*p*-Value
YOLOv3	13.41 ± 0.52	9.74 ± 0.38	<0.01 *
YOLOv4	9.23 ± 0.39	7.34 ± 0.36	<0.01 *
YOLOv5x	18.25 ± 0.62	12.80 ± 0.43	<0.01 *

Note: $\vec{G_{1}P_{1}}$, the distance between the front points of the ground-truth (***G*_1_**) and YOLO model prediction (***P*_1_**); $\vec{G_{2}P_{2}}$, the distance between the rear points of the ground-truth (***G*_2_**) and YOLO model prediction (***P*_2_**). The value coordinates in this calculation were in pixels; *, a significant difference (*p* < 0.01).

## Data Availability

The data used to support the findings of this study are available from the corresponding author upon request.

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
