# Peer review of "A Deep Learning Method for Foot Progression Angle Detection in Plantar Pressure Images"

_sensors, 2022, doi:10.3390/s22072786_

Round 1

Reviewer 1 Report

In this paper, a foot progression angle detection method based on YOLO network in plantar pressure images is proposed. The results on the public dataset are presented to analyze the detection performance. However, this paper is not well-structure, and its style is poor. A major rewritten is needed before publication.

  1. The title of this paper is not suitable, as the goal of this paper is to present a FPA detection method, but not a system. Please modify the title.

  1. The introduction part is not complete and must be enriched. You must add the following descriptions and references.

There are two strategies to capture the information of pressure patterns and distributions, e.g., foot progression angle. The first one is the plantar pressure signal; it is one-dimensional signal. The introduction of plantar pressure signal can be referred to [1-2]; lots of signal processing method [3-4] can be used for plantar pressure signal analysis.

[1] Liu, Wei et al. “Plantar Pressure Detection System Based on Flexible Hydrogel Sensor Array and WT-RF.” Sensors (Basel, Switzerland) 21 (2021).

[2] M. Hagan and H. -N. Teodorescu, "Sensors for Foot Plantar Pressure Signal Acquisition," 2021 International Symposium on Signals, Circuits and Systems (ISSCS), 2021, pp. 1-4, doi: 10.1109/ISSCS52333.2021.9497425.

[3] "Target Detection Within Nonhomogeneous Clutter Via Total Bregman Divergence-Based Matrix Information Geometry Detectors," in IEEE Transactions on Signal Processing, vol. 69, pp. 4326-4340, 2021.

[4] H. Li, F. Wang, C. Zeng and M. A. Govoni, "Signal Detection in Distributed MIMO Radar With Non-Orthogonal Waveforms and Sync Errors," in IEEE Transactions on Signal Processing, vol. 69, pp. 3671-3684, 2021, doi: 10.1109/TSP.2021.3087897.

The other one is the plantar pressure images; it is two-dimensional signal.

Obviously, the two-dimensional image is much more complex than the one-dimensional signal. Why don’t you employ the plantar pressure signal to get the foot progression angle information? Please specify.

  1. At the end of the introduction part, please add the outline of this paper.

  1. The symbols are not bold in the equations, please keep consistent with the text.

  1. It is not very convinced that you just provide the results of YOLO network on the FPA detection performance. You should also provide some results of other detectors, e.g., RCNN, Fast-RCNN, SSD, or give more descriptions on why you only use the YOLO network.

  1. You should add a Section to introduce the frameworks of deep learning detection methods.

Author Response

Thank you very much for your thoughtful review of our manuscript. We are grateful to the reviewers for the insightful comments on our paper. We have been able to incorporate changes to reflect most of the suggestions provided as attached. 

Reviewer 2 Report

The topic is one of importance given the high number of presentations to health services that are related to concerns on  
the prevalence and related factors with plantar pressures in the  foot problems population. Also, this is an interesting aim with  this study to aims to establish a precision model for determining the FPA, which is beneficial to reducing knee pain and important outcome associated with knee osteoarthritis progression. I think it would be a more clear study if the following parts were revised and supplemented. These will be discussed below relative to the information of the manuscript.

General Comments:
Overall the manuscript is generally well written and is a topic of interest. However after reading it a number of times I am still left without key take-home points. I believe these points are in the results they just need to be developed.

Specific comments:
Abstract:
1) The authors state they were to establish a precision model for determining the FPA, which is beneficial to reducing knee pain and important outcome associated with knee osteoarthritis progression. This seems like too much of an over simplification of what was done. I do feel that it would be beneficial to explain what specifically you are looking at in relation to plantar pressures injury (this also applies to the main body of the paper). Is it the development of FPA injury literature. This needs to be made clearer throughout the paper. (Major Compulsory Revision)

Introduction
2) The first paragraph should have a sentence or two added that better outlines why this study is important related with the impact of FPA in chronic disease and too Dynamic Parameters of the Feet during Gait https://pubmed.ncbi.nlm.nih.gov/34834508/ 

Furthemore, the authors do a poor job on reviewing relevant literatura related with importance with changes suface and plantar pressure https://pubmed.ncbi.nlm.nih.gov/34834508/ 

3) In the last paragraph, the significance of the proposed word should be included highlighting why your work is important. what is the scientific contribution of this paper? it is not clear how this paper can make a significant contribution to the state of the art. (Major Compulsory Revision).
In addition, author´s hypotheses should be included and to change teh flow chart of the figure one for the method section.

5) This methods section is poor, needs to present a better rationale for the study and the methodology employed. Also, neither appear information related with inclusion and exclusion criteria, dates, protocol. The study design is a experimental research of ramdom sampling method, where the study was conducted in the hospital or in the university center? This research adhere to reporting STROBE guidelines? (Major Compulsory Revision).

6) Where the experiments carried out? In a hospital? In an educational institution? Provide this information. Also, update the link related with this plantar pressure images because not found this   download from https://aidea-web.tw/topic/d6d8b111-d915-4ea9-89ee-43e148c37f6e.

7) Add figure 1 as a study flow chart for the readers. (Major Compulsory Revision).
8) The Discussion section is a rehashing of the results. It does not appear that the authors include much interpretation of what the study findings mean for clinical practice or research. (Major Compulsory Revision)

FInally, the conclusión is weak and too long. (Major Compulsory Revision)

Author Response

Thank you very much for your thoughtful review of our manuscript. We appreciate the time and effort that the reviewer has dedicated to providing your valuable feedback on our manuscript. We are grateful to the reviewers for their insightful comments on this paper. We have incorporated changes to reflect most of the suggestions provided as attached.

Round 2

Reviewer 1 Report

The authors have addressed my concerns well.

Reviewer 2 Report

I think the authors addressed the concerns. It is ready for publication.